# LEMON - A FOUNDATION MODEL FOR SINGLE-CELL NUCLEAR MORPHOLOGIES FOR DIGITAL PATHOLOGY

## ABSTRACT

Representation learning is a central challenge in Computational Pathology (CP), with direct implications for cancer research and precision medicine. While Self-Supervised Learning (SSL) has advanced patch and slide-level analysis of Whole-Slide Images (WSIs), single-cell representation learning has remained under-explored, despite its importance for characterizing cell types and phenotypes. We introduce LEMON (Learning Embeddings from Morphology Of Nuclei), a self-supervised foundation model for scalable single-cell image representation. Trained on millions of cell images spanning diverse tissues and cancer types, LEMON provides versatile and robust morphology representations that enable large-scale single-cell studies in pathology. We demonstrate its effectiveness across diverse prediction tasks on five benchmark datasets, establishing LEMON as a new paradigm for cell-level computational pathology.

## 1 INTRODUCTION

The proper function of tissue depends on the coordinated interaction of vast numbers of cells of diverse types, whose spatial arrangement is essential for maintaining architecture and whose disruption often signals disease. The composition and interactions of these individual cells can strongly influence disease development, and in some cases predict treatment response, providing biomarkers of high clinical value. A prominent example is the Tumor Microenvironment (TME), where understanding the interplay between different cell types is critical for predicting disease progression, patient survival, and therapeutic response (de Visser & Joyce, 2023; Giraldo et al., 2019; Wang et al., 2023). Computational methods for the analysis of tissues at single-cell resolution are therefore important, both for mechanistic research and for clinical applications.

In recent years, numerous imaging technologies have enabled the molecular characterization of single cells in their spatial context, at either the transcriptional or protein level. However, these techniques are extremely costly, and datasets remain limited in scale. By contrast, Hematoxylin and Eosin (H&E) staining is the most widely used imaging modality in pathology. As a routine clinical practice, it provides detailed information on cell morphology and spatial organization, and large-scale datasets containing hundreds of thousands of images are available. This abundance has fueled the emerging field of computational pathology.

Learning robust feature representations from H&E images with limited labels remains a core challenge in computational pathology. Recent advances have largely been driven by SSL, which reduces dependence on manual annotation (Balestriero et al., 2023). Because WSIs can exceed 50,000 pixels in width, most SSL approaches operate at the patch level, learning embeddings from small tiles cropped from WSIs (Wang et al., 2022; Zimmermann et al., 2024; Saillard et al., 2024; Chen et al., 2024). More recent work integrates the aggregation of patch features within the SSL framework to produce a single slide-level representation (Lazard et al., 2022; Jaume et al., 2024b; Ding et al., 2024; Vaidya et al., 2025). While effective for slide-level predictions, these approaches prevent the analysis of fine-grained morphological details and cellular organisation. To date, work on cell-level representation learning remains sparse. Notable examples include DinoBloom, which was designed for hematology cell images from blood smears (Koch et al., 2024), and Volta (Nakhli et al., 2024), a self-supervised contrastive learning model trained on a limited set of tissue-derived cell images. These pioneering works are, however, limited by either their domain specificity or the scale and diversity of their training data.

A prerequisite for training powerful, general-purpose SSL models for cells is the assembly of a large-scale, diverse datasets of cell images from WSIs. Recent advances in nucleus segmentation have made large-scale cell extraction increasingly feasible. Models like Hover-Net (Graham et al., 2019) and CellViT (Hörst et al., 2023; 2025), along with highly optimized implementations (Liakopoulos et al., 2024), can now automatically segment millions of cells across various tissue types in a reasonable amount of time. This development opens the door to leveraging state-of-the-art SSL paradigms, such as DINOv2 (Oquab et al., 2024) and MoCov3 (Chen et al., 2021), which are known to benefit significantly from larger and more diverse training datasets both for natural images (Vo et al., 2024) and for pathology images (Chen et al., 2025).

In this work, we capitalize on these advancements to address the gap in cell-level representation learning. We introduce LEMON (Learning Embeddings from Morphology Of Nuclei), a new family of SSL models for cell images. Our work not only sets a new state of the art for cell representation learning but also provides a comprehensive blueprint for creating and training such models on this particular type of images. Our main contributions are:

- We introduce a family of models trained with several contrastive and non-contrastive SSL paradigms, to learn powerful representations of nuclear morphology.
- We curate several large-scale cell datasets derived from a diverse cohort of whole-slide images, and we systematically study the impact of dataset scale and composition (number of cells, slides, and organs) on downstream performance.
- We design and validate a set of domain-specific data augmentations tailored to the visual characteristics of H&E stained cell images, demonstrating their crucial role in model generalization.
- We conduct an extensive benchmark evaluation showing that LEMON significantly outperforms prior art on five distinct downstream tasks, encompassing both cell classification and regression challenges of varying difficulty.

## 1.1 SELF-SUPERVISED LEARNING

**Contrastive Learning.** The objective of contrastive learning is to learn an embedding space where different augmented views of the same image (a positive pair) are pulled together, while views from different images (negative pairs) are pushed apart. This is often formalized using the InfoNCE loss (Oord et al., 2019). Given a query view $q$ and a set of key views $\{k_0, k_1, ..., k_N\}$, where $k_0$ is the positive key corresponding to $q$ and $\{k_i\}_{i=1}^N$ are negative keys, the loss is defined as:

$$\mathcal{L}_q = -\log \frac{\exp(\text{sim}(q, k_0)/\tau)}{\sum_{i=0}^N \exp(\text{sim}(q, k_i)/\tau)}$$

Here, $\text{sim}(\cdot, \cdot)$ is a similarity function (e.g., cosine similarity) and $\tau$ is a temperature hyperparameter. While early methods struggled with strategies for sourcing effective negative pairs (Xu et al., 2022), modern approaches like **MoCov3** (Chen et al., 2021) have streamlined this process. MoCov3 simplifies its predecessors by using a large batch of in-flight samples for the negative keys, removing the need for a dedicated memory bank. It retains the teacher-student framework, where the student network (which produces the query $q$) is trained via backpropagation, and the teacher network (which produces the key $k$) is an Exponential Moving Average (EMA) of the student's weights. This momentum-based update ensures the keys remain consistent, which is crucial for stable training.

**Non-Contrastive Learning.** Non-contrastive methods learn representations by matching the outputs for different views of the same image, thereby avoiding the need for explicit negative pairs. To prevent representational collapse, they employ a teacher-student architecture where the teacher's weights $\theta_t$ are an EMA of the student's weights $\theta_s$. The state-of-the-art **DINOv2** (Oquab et al., 2024) model advances this paradigm by composing three distinct loss functions: an image-level matching loss (from DINO) (Caron et al., 2021), a patch-level masked-image modeling loss (from iBOT) (Zhou et al., 2022), and a regularization loss (KoLeo).

The core objective, inherited from **DINO** (Caron et al., 2021), operates on multiple views of an image generated by a multi-crop augmentation strategy. This strategy produces a set of high-resolution "global" views ($\mathcal{V}_g$) and a set of lower-resolution "local" views ($\mathcal{V}_l$). The teacher network processes only the global views, while the student network processes all views. The goal is for the student's

output distribution $P_s$ to match the teacher's sharpened and centered distribution $P_t$. The loss is composed of two cross-entropy terms:

- **Global-to-Global Matching:** The student's output for one global view is matched against the teacher's output for the other global view.

$$\mathcal{L}_{g \leftrightarrow g} = - \sum_{\substack{v,v' \in \mathcal{V}_g \\ v \neq v'}} P_t(v') \log P_s(v)$$

- **Local-to-Global Matching:** The student's output for each local view is matched against the teacher's output for all global views.

$$\mathcal{L}_{l \leftrightarrow g} = - \sum_{v_l \in \mathcal{V}_l} \sum_{v_g \in \mathcal{V}_g} P_t(v_g) \log P_s(v_l)$$

The total image-level loss is $\mathcal{L}_{\text{DINO}} = \mathcal{L}_{g \leftrightarrow g} + \mathcal{L}_{l \leftrightarrow g}$.

To encourage the learning of fine-grained local features, DINOv2 integrates the masked-image modeling objective from **iBOT** (Zhou et al., 2022). For an input image, a random subset of its patches $\mathcal{M}$ is masked. The teacher receives the full, unmasked image, while the student receives the masked image. The student is then tasked with predicting the teacher's output tokens for the masked patches. The loss is a cross-entropy between the student's predictions and the teacher's distribution for the masked patches:

$$\mathcal{L}_{\text{iBOT}} = - \sum_{x_m \in \mathcal{M}} P_t(x_m) \log P_s(x_m)$$

To further prevent feature collapse and promote feature space uniformity, DINOv2 adds the **KoLeo** regularizer. This loss acts on the batch of embeddings produced by the teacher network. It encourages diversity by penalizing similarity between distinct images, specifically by maximizing the distance between an embedding $z_i$ and its nearest neighbor $z_{NN(i)}$ in the batch. The loss is formulated as:

$$\mathcal{L}_{\text{KoLeo}} = \sum_{z_i \in \text{batch}} - \log \|z_i - z_{NN(i)}\|_2^2$$

This pushes embeddings apart, ensuring the feature space does not collapse to a small volume. The final DINOv2 loss is a weighted sum of these three components: $\mathcal{L} = \mathcal{L}_{\text{DINO}} + \alpha \mathcal{L}_{\text{iBOT}} + \beta \mathcal{L}_{\text{KoLeo}}$.

## 1.2 SSL in Histopathology

The paradigms of self-supervised learning, proven effective on natural images, have been widely adapted for computational pathology, predominantly at the level of image tiles or patches. These methods can be broadly categorized by their core SSL strategy and their operational scale.

**Tile-Level Representation Learning.** Early successes in pathology SSL often involved adapting contrastive learning frameworks. For instance, **CTransPath** (Wang et al., 2022) modified the MoCo framework by defining positives not only as augmented views of the same tile, but also as other tiles with high cosine similarity, which reflects the idea that tiles are much more similar to each other than natural images. More recently, the field has shifted towards building non contrastive methods. Inspired by the success of DINOv2, models like **UNI** (Chen et al., 2024), **H-Optimus-0** (Saillard et al., 2024), and **Virchow2** (Zimmermann et al., 2024) have demonstrated the benefits of pre-training Vision Transformers on massive, curated datasets comprising millions of histopathology tiles. The primary contribution of these works lies in meticulous data curation and demonstrating remarkable scaling laws, where model performance on diverse downstream tasks consistently improves with the scale of the pre-training data.

**Cell-Level Representation Learning.** In contrast to the extensive work at the tile level, research on single-cell level representations remains limited. To our knowledge, **Volta** (Nakhli et al., 2024) is the only SSL model specifically designed for this task in the context of histopathology. Volta introduces a novel dual-contrastive learning objective. The first objective is a standard instance-discrimination task, where augmented views of the same cell are pulled together in the embedding space. The second objective contrasts the cell's representation against a representation of its local

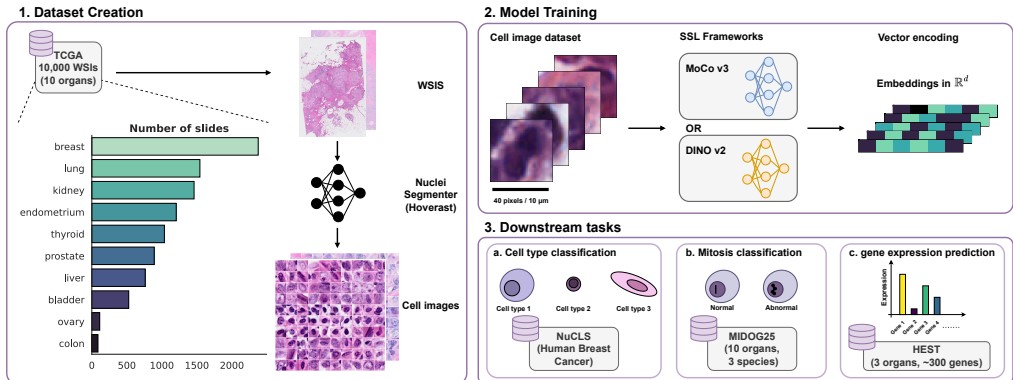

Figure 1: Overview of the LEMON framework.

tissue microenvironment, which is obtained from the surrounding background patch after masking cell nuclei.

## 2 METHODS

### 2.1 A LARGE-SCALE CELL DATASET

To train our models, we generated a large-scale dataset of cell images from The Cancer Genome Atlas (TCGA) database (Weinstein et al., 2013). We processed around 10,000 WSIs scanned at 40X resolution spanning 10 distinct cancer types. To perform cell segmentation at this scale, we employed **HoverFast** (Liakopoulos et al., 2024), a highly optimized implementation of the Hover-Net architecture that provides a 20x speed-up, making slide-level segmentation computationally tractable. For each detected nucleus, we extracted a $40 \times 40$ pixel patch centered on the nucleus centroid. After that we selected random nuclei from each WSI. This process yielded a pre-training dataset of several hundred million cell instances. The small patch size, a key characteristic of this domain, starkly contrasts with the larger resolutions (e.g., $224 \times 224$) common in natural image, necessitating adaptations in both model architecture and training methodology.

### 2.2 SSL FOR CELL IMAGES

We adapted and compared the two leading SSL paradigms, contrastive learning (**MoCov3** (Chen et al., 2021)) and non-contrastive learning (**DINOv2** (Oquab et al., 2024)), tailoring them to the specific properties of our cell image data.

**Architectural and Framework Adaptations.** Given the small input resolution of our cell patches, we utilized smaller model backbones than those typically used for SSL in natural images. For the contrastive approach with MoCov3, we experimented with Vision Transformer (ViT) (Dosovitskiy et al., 2021), ViTs refers to the *small* architecture and ViTb to the *base* architecture. We used a patch size of 8 in our experiments which we indicate with ViTs8 for a *small* architecture. For the DINOv2 framework, we made two critical modifications to the loss function. First, we removed the local-to-global matching component of the DINO loss, as local crops of an already small $40 \times 40$ patch carry insufficient information to be meaningful. Second, consistent with observations in other pathology applications (Zimmermann et al., 2024), we found the effect of the KoLeo regularizer to be excessively *strong*, potentially leading to numerical errors in the loss. We therefore replaced it with a **more moderate** Kernel Density Estimation (KDE) based loss, which encourages global feature uniformity without excessively penalizing local neighborhoods in the embedding space:

$$\mathcal{L}_{\text{KDE}} = \sum_z \log \sum_{z'} k(z, z')$$

where $k$ is a kernel function measuring similarity between embeddings.

**Domain-Specific Data Augmentations.** We designed two custom augmentations to address the unique characteristics of cellular images. First, to enforce rotational invariance, a critical prior for cell morphology, which has no canonical orientation, we implemented a full **random rotation** from $0°$ to $360°$ similar to (Alfasly et al., 2024). To avoid edge artifacts (e.g., black corners), we implemented this by extracting a larger $60 \times 60$ patch, performing the rotation, and then taking a central $40 \times 40$ crop. Second, we evaluated a **random stain augmentation** (Shen et al., 2022) based on Macenko's method (Macenko et al., 2009). This simulates the variability in H&E staining that arises from different slide preparation protocols, training the model to learn features that are invariant to color and intensity shifts and thus more robust for downstream applications.

## 2.3 BENCHMARKING DATASETS

To comprehensively evaluate the quality of the learned cell representations, we benchmarked our models on eight distinct downstream tasks, encompassing five classification tasks and three regression tasks. These benchmarks were selected to test various aspects of the embeddings, including fine-grained cell type discrimination, generalization across different domains, and the ability to predict molecular features from morphology.

- **NuCLS** (Amgad et al., 2022): A multi-class classification dataset of single-cell images from breast cancer H&E slides. Its key feature is a hierarchical annotation scheme, with labels reviewed and confirmed by pathologists. This structure allows us to evaluate the representative power of our embeddings on both fine-grained (e.g., distinguishing immune cell types) and coarse-grained (e.g., tumor vs. stromal) classification tasks. We refer to them as NuCLS (super) for broad categories, NuCLS (main) for main cell type categories and NuCLS (raw) for fine-grained type and subtype classifications. It is derived from TCGA WSIs and can be considered an in-domain dataset for LEMON.

- **PanNuke** (Gamper et al., 2020): A large-scale, multi-organ dataset of segmented and labeled nuclei from diverse tissue types. It contains a broad range of nuclear morphologies and cell types across multiple cancer and non-cancer tissues, making it well suited for evaluating generalization across organs and histological contexts. Here, we use PanNuke for single-cell classification, assessing the robustness of learned representations in a heterogeneous, multi-domain setting. It is primary an out-of-domain (OOD) dataset for LEMON, although some images may originate from TCGA.

- **MIDOG25** (track 2, training dataset)(Weiss et al., 2025): A dataset for binary classification of mitotic figures (atypical vs. normal). It is specifically designed to test model robustness and generalization, as it contains cell images sourced from multiple species (human and canine), diverse organs, and various digital slide scanners. It an OOD dataset for LEMON.

- **HEST-derived Gene Expression Regression** (Jaume et al., 2024a): To assess the capacity of our embeddings to capture morpho-molecular relationships, we curated a novel regression dataset from the HEST database. We created 3 datasets by using slides originating from breast cancer (HEST-breast), lung pulmonary fibrosis (HEST-lung) and bowel cancer (HEST-bowel). It an OOD dataset for LEMON.

Further details on each dataset and preprocessing are provided in the subsection A.1.

## 2.4 EMBEDDING EVALUATION

To assess the quality and discriminative power of the features learned by our SSL models, we followed the standard **linear probing** protocol. This evaluation involves training a simple linear model on fixed embeddings obtained from the pretrained model to perform a specific downstream task. The rationale for this approach is that a powerful feature extractor should produce representations that are linearly separable. Consequently, the performance of a simple linear model serves as a direct and reliable proxy for the quality of the learned embeddings. Alternative protocols like full fine-tuning, parameter-efficient fine-tuning or few shot learning also exist, but these methods generally produce similar conclusions about relative model performance (Marks et al., 2024). In our experiments, we trained a **logistic regression** classifier on the embeddings for the classification tasks (NuCLS, PanNuke and MIDOG25) and a **ridge regression** model for the gene expression prediction task (HEST).

The performance of these linear models is used to directly compare the efficacy of our different SSL training configurations.

Second, we conducted an additional comparative analysis to assess how different representations behave under domain shift. To this end, we simulated staining variations Macenko et al. (2009), and evaluated the changes induced in the latent space across the different SSL strategies. In addition, we measured the impact of this staining-induced domain shift on downstream classification performance on NuCLS and MIDOG25.

Third, we evaluated how well our embeddings capture biological differences by comparing the morphology-driven structure of the latent space with the expression patterns of well-known marker genes.

## 3    RESULTS

### 3.1    EXPERIMENTAL SETUP

Unless stated otherwise, we used a dataset of 10M cell images from 10 organs. All the models using the MoCo v3 pretraining strategy are trained for 150 epochs with an epoch length of 1M images. The batch size is set to 4096. For the augmentations, we used the best combination that we found as explained later in the experiments.

### 3.2    FULL BENCHMARKING

| Embeddings | MIDOG25 bal acc | NuCLS bal acc | HEST-Bowel pcc | HEST-Breast pcc | HEST-Lung pcc |
|---|---|---|---|---|---|
| *Natural images TL* | | | | | |
| Imagenet (ResNet-18) | 0.577 (0.008) | 0.693 (0.005) | 0.114 (0.023) | 0.113 (0.023) | 0.083 (0.021) |
| Imagenet (ResNet-50) | 0.604 (0.013) | 0.712 (0.005) | 0.12 (0.035) | 0.124 (0.023) | 0.091 (0.013) |
| *Natural images FM* | | | | | |
| DINOv2 (ViT-s/14) | 0.642 (0.004) | 0.732 (0.004) | 0.146 (0.016) | 0.157 (0.029) | 0.116 (0.028) |
| DINOv2 (ViT-b/14) | 0.653 (0.013) | 0.732 (0.004) | 0.145 (0.022) | 0.156 (0.026) | 0.118 (0.028) |
| DINOv2 (ViT-l/14) | 0.645 (0.008) | 0.721 (0.006) | 0.142 (0.016) | 0.153 (0.026) | 0.117 (0.029) |
| DINOv2 (ViT-g/14) | 0.654 (0.007) | 0.734 (0.009) | 0.153 (0.021) | 0.164 (0.025) | 0.118 (0.029) |
| *Tile level FM* | | | | | |
| UNI (ViT-l/16) | 0.619 (0.007) | 0.733 (0.006) | 0.15 (0.035) | 0.151 (0.029) | 0.118 (0.029) |
| Virchow2 (ViT-h/14) | 0.621 (0.011) | 0.754 (0.007) | 0.148 (0.034) | 0.154 (0.029) | 0.11 (0.029) |
| H-Optimus-0 (ViT-g/14) | 0.631 (0.013) | 0.751 (0.006) | 0.159 (0.029) | **0.157** (0.029) | 0.119 (0.029) |
| *Cell level FM* | | | | | |
| Volta (ResNet-18) | 0.726 (0.01) | 0.736 (0.003) | 0.148 (0.025) | 0.115 (0.028) | 0.118 (0.025) |
| LEMON-MoCov3 (ViT-s/8) | **0.746** (0.015) | **0.779** (0.005) | **0.16** (0.027) | 0.156 (0.021) | **0.128** (0.028) |

Table 1: Performance of pretraining strategy on downstream tasks. We report balanced accuracy for NuCLS (supervised) and MIDOG25 (majority), and Pearson Correlation Coefficient for HEST (breast, lung, bowel). Values are mean (s.d.) over folds, best values are highlighted in bold.

To assess performance, we benchmarked our model against strong baselines that use pre-trained representations, grouped into three families: (1) models pre-trained on natural images: transfer learning from ImageNet (Russakovsky et al., 2015) and the self-supervised DINOv2 (Oquab et al., 2024); (2) foundation models trained on histopathology tile images from WSIs, including UNI (Chen et al., 2024), H-Optimus-0 (Saillard et al., 2024), and Virchow2 (Zimmermann et al., 2024); and (3) Volta (Nakhli et al., 2024), the current state of the art for cell-level representations. For transfer learning (Zhuang et al., 2020), we extracted embeddings from each model's penultimate layer. Because most baselines (except Volta) require 224×224 inputs, we resized nuclei images to the correct size. Results are reported in Table 1.

Our method decisively outperformed models pre-trained on natural images across all datasets. On MIDOG25, for instance, it achieves a balanced accuracy of $0.746 \pm 0.015$, compared with

$0.604 \pm 0.005$ for an ImageNet-pretrained ResNet-50 and $0.654 \pm 0.007$ for the best DINOv2 variant. These gaps underline the limits of natural-image features and the importance of domain-specific pre-training for cellular morphology.

Against contemporary tile-level Foundation Model (FM)s (UNI, H-Optimus-0, Virchow2), our approach performs significantly better on NuCLS and MIDOG25, and remains highly competitive on the HEST gene-expression prediction tasks. Gains are particularly pronounced on MIDOG25, which depends on fine-grained cellular cues that tile-level FMs tend to overlook. In order to check that this results is not entirely due to domain shift introduced by feeding nucleus images to tile-level FM we made an experiment with the MIDOG25 dataset, where we used different strategies to extract cell-level encodings from tile-level FM applied to larger fields of view matching the FM training distribution. Our approach also outperforms these tailored encoding strategies (for more details, see subsection A.6).

Compared to Volta, the only other cell-level baseline, our model held a clear advantage across all tasks. It surpassed Volta on NuCLS and MIDOG25 classification and delivered stronger results on HEST gene-expression prediction, indicating more robust and generalizable cellular representations for downstream computational pathology.

Additional evaluation metrics for NuCLS and MIDOG25, as well as results on the PanNuke dataset, are provided in the Supplementary Material (see subsection A.7).

Finally, we also investigated the stability of our latent space under domain shifts. For this, we simulated staining differences Macenko et al. (2009) and investigated the induced differences in the latent space, both by investigating the differences at the representation level and the impact on downstream classification tasks under domain shift. LEMON demonstrated greater robustness compared to H-Optimus-0 and Volta (see Table 7 and Figure 7).

### 3.3 FLOATING POINT OPERATIONS (FLOPs) COMPARISON

Computational efficiency is crucial in the single-cell setting, where a single whole-slide image may contain millions of nuclei, making large-scale embedding extraction prohibitively expensive with heavyweight architectures. Tile-level foundation models typically rely on very large backbones (e.g., Vi-H/14, ViT-g/14) with high computational footprints, whereas LEMON leverages a lightweight ViT-s/8 backbone, operating with orders of magnitude fewer FLOPs.

To evaluate the trade-off between performance and computational cost, we compared different pre-training strategies by plotting downstream performances against the number of FLOPs required during inference (Figure 2). Our cell-level models achieves state-of-the-art performance while striking an optimal balance between representational power and efficiency. Interestingly, for classification tasks, histopathology tile-level models generally show improved performance with increasing model size or FLOPs. In sharp contrast, LEMON breaks this trend, achieving the highest accuracy overall (NuCLS super: 0.773, MIDOG25: 0.748) with only 0.5G FLOPs, compared to 295.9G FLOPs for H-Optimus-0, the second-best performing model. For gene expression prediction, a more challenging task, the overall trends remain consistent.

### 3.4 CHOICE OF SSL FRAMEWORK

We compared two state-of-the-art SSL methods, MoCov3 Chen et al. (2021) and DINOv2 Oquab et al. (2024), as pretraining strategies for small histopathology patches (see Table 6). This choice was driven by the central role of these two SSL-methods in existing tile-level pathology foundation models. While DINOv2 is one of the strongest SSL frameworks available, we observed several limitations when applied to small 40×40 images. To address these, we introduced two modifications to the loss function: (i) removing the local-to-global matching term, which is ill-suited for such small patches, and (ii) replacing the strong KoLeo regularizer with a milder KDE-based loss. These two modifications improved performance across all classification benchmark datasets. Despite these adjustments, MoCov3 consistently outperformed DINOv2, suggesting that contrastive SSL is more robust in low-resolution pathology scenarios. We hypothesized that this advantage arises from the limited image size, which diminishes the effectiveness of DINO-style local/global matching while favoring the global contrastive objective used by MoCov3. Consequently, we focused the remainder of the study on MoCov3, and LEMON refers to LEMON-MoCov3.

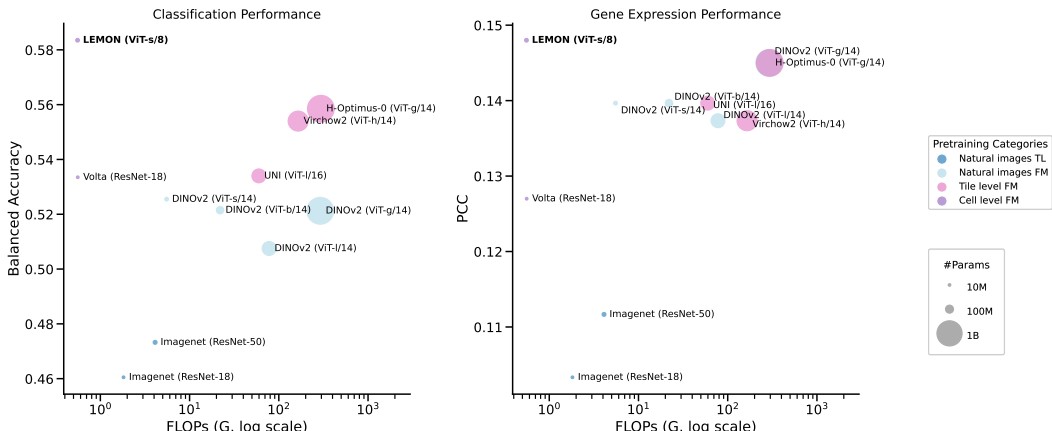

Figure 2: **Performance of pretraining strategies versus FLOPs.** Left: mean balanced accuracy across NUCLS (super, main, raw) and MIDOG25 for classification. Right: mean Pearson Correlation Coefficient across breast, lung, and bowel datasets for regression. Marker size indicates model parameter count; color indicates pretraining category. Cell-level foundation models achieve superior performance compared to both natural image pretraining and tile-level foundation models, while requiring significantly fewer FLOPs.

## 3.5 DATASET COMPOSITION

We first examined how dataset scale affects benchmark classification. As shown in the top row of Figure 3, balanced accuracy increases consistently as the dataset grows, before saturating around 1M images for both ViT-S/8 and ViT-B/8. We kept ViT-S/8 as increasing the model sizes did not lead to significantly better results with our training scheme.

We then isolated the role of data diversity, motivated by evidence that training set diversity can strongly influence downstream performance (Oquab et al., 2024; Vo et al., 2024). Fixing the dataset size at 1M images, we varied the number of WSIs and the number of source organs used to extract cell images. As shown in the bottom row of Figure 3, gains are driven primarily by increasing the number of slides, while performance remained relatively stable with respect to the number of organs, provided that the number of slides is sufficiently high. This suggests the model does not require organ-specific pretraining for effective application across organs.

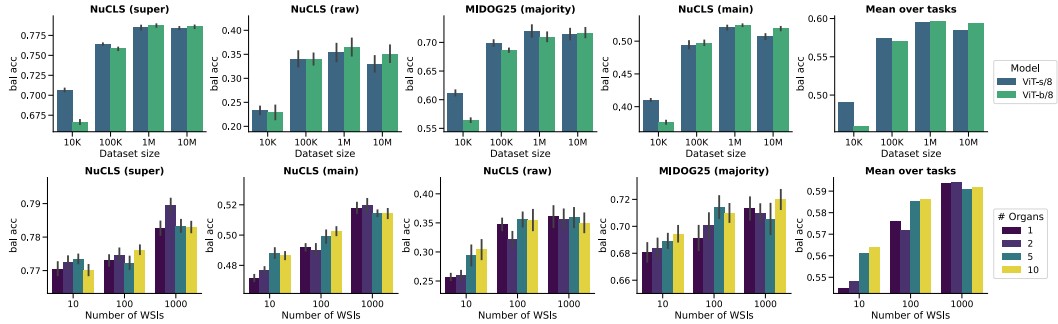

Figure 3: **Performance of models vs. training data composition.** Mean balanced accuracy (error bars represents standard errors) across MIDOG25 and the three NuCLS classification tasks. *Top:* Performance by training dataset size; models improve with larger datasets and level off near 1M images. *Bottom:* With the total number of images held constant, models perform better as diversity increases (more organs of origin and more slides).

## 3.6 DOMAIN-SPECIFIC AUGMENTATIONS

Augmentation policy strongly affects SSL performance (Morningstar et al., 2024). We revised the pipeline with stronger color jitter, random patch masking, and a customized rotation protocol without padding (`a1`), which improved both classification (NuCLS, MIDOG25) and gene-expression prediction (HEST) with respect to the standard augmentation scheme (`a0 (MoCo)`).

Imposing robustness to color variations is important in digital pathology, as staining protocols and scanners may vary between hostpitals. Adding grayscale augmentations (`a1+gray`) (Kang et al., 2023) yielded the best MIDOG25 score and competitive NuCLS results, though HEST effects were mixed. Surprisingly, RandstainNA color perturbations (Shen et al., 2022) mimicking realistic color variations helped HEST when using a single color template (`a1+gmm1`) but reduced classification accuracy, and the multi-template variant (`a1+gmm10`) underperformed overall.

Balancing these trade-offs, we adopted `a1+gray` as our default, as it combines the strongest classification gains with good HEST performance. The results are summarized in Table 2.

Table 2: Effect of augmentation policy on downstream tasks. We report balanced accuracy for NuCLS (supervised) and MIDOG25 (majority), and Pearson Correlation Coefficient for HEST (breast, lung, bowel). Values are mean (s.d.) over folds, best values are highlighted in bold.

| | MIDOG25 bal acc | NuCLS bal acc | HEST-Bowel pcc | HEST-Breast pcc | HEST-Lung pcc |
|---|---|---|---|---|---|
| Augmentations | | | | | |
| a0 (MoCo) | 0.682 (0.013) | 0.771 (0.003) | 0.154 (0.03) | 0.148 (0.023) | 0.116 (0.027) |
| a1 | 0.715 (0.021) | **0.784** (0.003) | 0.161 (0.028) | 0.155 (0.025) | **0.133** (0.032) |
| a1+gray | **0.746** (0.015) | 0.779 (0.005) | 0.16 (0.027) | 0.156 (0.021) | 0.128 (0.028) |
| a1+gmm1 | 0.736 (0.018) | 0.768 (0.005) | **0.164** (0.019) | **0.162** (0.021) | 0.126 (0.029) |
| a1+gmm10 | 0.725 (0.014) | 0.765 (0.003) | 0.159 (0.021) | 0.152 (0.022) | 0.123 (0.028) |

## 3.7 REPRESENTATION ANALYSIS

We further examined the learned cell embeddings to assess whether they capture meaningful biological structure. For this, we projected the embeddings into two dimensions using t-distributed Stochastic Neighbor Embedding (t-SNE) (Figure 4).

The spatial organization of the latent space revealed clear biological structure. Along the first dimension, cells located on the far left (1) were predominantly epithelial. Within this region, we identified a subpopulation in the central left area (2) characterized by large, rounded nuclei with irregular borders and prominent nucleoli, suggestive of neoplastic epithelial cells. In the upper right corner (3), we observed clusters of cells with small, round nuclei, consistent with lymphocytes. An adjacent cluster (4) likely corresponded to plasma cells, with eccentric nuclei. Finally, the lower right region (5) contained elongated, spindle-shaped nuclei, suggestive of fibroblasts or myofibroblasts.

These observations highlight that the model captures not only broad cell-type distinctions but also finer-grained morphological and pathological variations directly from the learned representations.

Furthermore, we investigated how the structure of our latent space correlates with gene expression, as a proxy for biological similarity of cells. For this, we examined the alignment between predicted marker-gene expression and the t-SNE embeddings on a Xenium slide. Even without any training on gene expression, certain patterns were localized to specific regions of the morphological embedding space, confirming that the learned embeddings capture biologically meaningful signals (see Figure 8).

## 4 DISCUSSION

In this paper, we introduce LEMON, a self-supervised model for nuclei image representation learning. We build a large-scale dataset of nucleus crops using segmentation networks and show that our

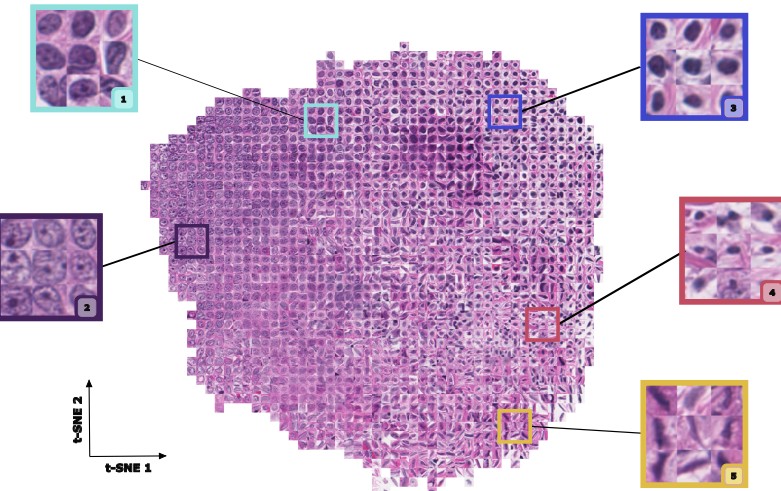

Figure 4: **Morphological map of learned cell representations.** A t-SNE projection of 100,000 cells randomly sampled from a breast cancer whole-slide image (TCGA) is shown. A regular grid of cell images visualizes the embedding space, revealing a smooth continuum of latent morphologies. Five representative regions were annotated by an expert pathologist, corresponding to distinct cellular phenotypes: (1) epithelial cells, (2) neoplastic epithelial cells with large, irregular nuclei, (3) small lymphocytes, (4) plasma cells, and (5) fibroblasts.

approach achieves state-of-the-art performance across five downstream tasks. Notably, we demonstrate that self-supervised learning can be effectively applied to very small images (40×40) with minor adaptations to existing frameworks. By carefully tuning training schemes and augmentations, LEMON produces stable and meaningful embeddings that capture nuclear morphology and chromatin organization, reducing the need for manual labels and enabling new explorations of the morphological landscape of cell images. Furthermore, we show that our representations are robust with respect to staining-induced domain shifts, and we demonstrate that our representations truly capture biological signal, as they correlate with the expression of well-known marker genes. Finally, we make the dataset of all nuclei detection, the pretext task targets and our foundation model publicly available to the scientific community. Beyond CP, these results suggest that SSL could be extended to other domains where only small image patches are available.

In future work, it would be interesting to further quantify batch effects across staining protocols and microscopes of our models to ensure robust generalization, an issue already documented for tile level FM in CP (Kömen et al., 2024). Another interesting avenue would be to compare representations learned at different physical scales (nucleus, tissue and slide level views) to best combine the information learned at different physical scales. Finally, our representations provide an excellent starting point for in-depth spatial modeling and tissue architecture analysis. We deliberately chose not to incorporate the spatial distribution of cells into this foundation model, so that future work on spatial analyses can remain disentangled from single-cell morphological representations, which we consider an important asset for future developments.

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

# A APPENDIX

## A.1 BENCHMARKING DATASETS

**NuCLS dataset:** It is a dataset of images from slides originating from breast cancer of annotated cell type images. It is composed of three classification tasks with increased cell type resolution *super*, *main*, *raw*. The label distribution is shown in Table 3.

(a) NuCLS (super)

| Cell type | Number of images |
|---|---|
| tumor_any | 16052 |
| sTIL | 13642 |
| nonTIL_stromal | 7613 |

(b) NuCLS (main)

| Cell type | Number of images |
|---|---|
| tumor_nonMitotic | 15874 |
| lymphocyte | 9617 |
| nonTILnonMQ_stromal | 6634 |
| plasma_cell | 4025 |
| macrophage | 979 |
| tumor_mitotic | 178 |

(c) NuCLS (raw)

| Cell type | Number of images |
|---|---|
| tumor | 15874 |
| lymphocyte | 9617 |
| fibroblast | 6258 |
| plasma_cell | 4025 |
| macrophage | 979 |
| ductal_epithelium | 399 |
| vascular_endothelium | 376 |
| apoptotic_body | 293 |
| mitotic_figure | 178 |
| neutrophil | 35 |
| myoepithelium | 33 |
| eosinophil | 2 |

Table 3: NuCLS dataset.

**MIDOG25 dataset:** It is a dataset of images from slides originating from multiple organs and species with mytotic figure annotations. We didn't use any of the metadata to create our splits except for labels to generate stratified splits. The label distribution is shown in Table 4.

**PanNuke dataset:** It is a dataset of images from slides originating from multiple organs and species with broad cell type annotations. We extracted cell images for cells whose associated images of size 60 by 60 fall within the full crops with the annotations to make sure that all the part of the cell is in the images. It resulted in the following dataset. The dataset was originally published in Gamper et al. (2020), we downloaded it from `https://huggingface.co/datasets/RationAI/PanNuke`. The classes corresponds to

1. Neoplastic
2. Inflammatory
3. Connective
4. Dead
5. Epithelial

**HEST dataset:** This dataset is built from Xenium slides, a technology for Imaging-based Spatial Transcriptomics (IST) that measures up to hundreds of distinct messenger RNA (mRNA) species while preserving their spatial localisation. After cell segmentation, we construct cell–count matrices in which each row corresponds to a cell, each column to a gene, and each entry to the number of transcripts detected in that cell. We use the segmentation provided by the Xenium software. The platform also provides a co-registered H&E slide aligned with the cell–count matrix, enabling extraction of cell images that are spatially aligned with the transcriptomic measurements.

The prediction task is to identify the top 50 highly expressed genes, defined as follows. For every slide, we compute the mean expression per gene, rank genes within the slide, combine the ranks across slides, and retain the 50 highest-ranked genes. To standardize sample size, we subsample

Table 4: MIDOG25 dataset.

| Class | Number of images |
|---|---|
| NMF | 10191 |
| AMF | 1748 |

Table 5: PanNuke dataset.

| Class | Number of images |
|-------|------------------|
| 0 | 39222 |
| 1 | 26108 |
| 2 | 17258 |
| 3 | 13457 |
| 4 | 1582 |

10 000 cells per slide uniformly at random. We report performances with a leave-one-out slide evaluation strategy.

We accessed the Xenium slides through the HEST database (Jaume et al., 2024a). The slide IDs used for each subset are:

- **HEST-bowel**: TENX149, TENX148, TENX147, TENX139, TENX111

- **HEST-breast**: NCBI783, NCBI785, TENX94, TENX95, TENX98

- **HEST-lung**: NCBI856, NCBI857, NCBI858, NCBI859, NCBI860, NCBI861, NCBI864, NCBI865, NCBI866, NCBI867, NCBI870, NCBI873, NCBI875, NCBI876, NCBI879, NCBI880, NCBI881, NCBI882, NCBI883, NCBI884

We note that LEMON was pretrained on whole-slide images originating from TCGA. Consequently, NuCLS, which is derived from TCGA WSIs, can be considered an in-domain dataset for LEMON. PanNuke aggregates data from multiple sources, including TCGA. Therefore, it can be seen primarily as an OOD dataset for LEMON, although a subset of the data may be in-domain. Conversely, MIDOG25, and HEST do not overlap with LEMON's pretraining sources and can therefore be seen as OOD.

## A.2 COMPARISON OF SSL METHODS

We evaluated two state-of-the-art SSL methods, MoCo v3 and DINOv2, on small cell histopathology patches (40×40). As discussed in subsection 2.2, we tested several variants of DINOv2, including the original baseline and modified versions where (i) the local-to-global matching term was removed, and (ii) the KoLeo regularizer was replaced with a KDE-based loss.

The KDE loss employs a von Mises–Fisher kernel:

$$k_{vMF}(x, y) = exp(\kappa x^T y)$$

where $\kappa$ controls the concentration of the kernel around the mean direction. In our experiments, we set $\kappa$ to 5 and KDE loss weight to 0.05. This encourages global feature uniformity while avoiding excessive penalties on local neighborhoods.

Table 6 summarizes the performance of MoCo v3 and DINOv2 variants across the five benchmark datasets. MoCov3 consistently achieves the best results, outperforming all DINOv2 variants on Nu-CLS (0.779 vs 0.767), MIDOG (0.746 vs 0.670–0.678), and HEST Bowel (0.160 vs 0.117–0.142), which demonstrates its robustness for small 40×40 patches. The ablations on DINOv2 indicate incremental improvements: removing the local-to-global matching term slightly improves performances on most dataset (e.g,: Nucls: 0.703 vs 0.681, HEST-Breast 0.129 vs 0.119). Replacing the KoLeo regularizer with the KDE-based loss yields the best DINOv2 variant, improving NuCLS (0.767), HEST Breast (0.143), HEST Lung (0.147) and HEST Bowel (0.142) relative to the baseline.

## A.3 AUGMENTATION STRATEGIES

We here detail the augmentation strategies that we used to train our models. For MoCov3, we used the augmentations that we found in the pytorch implementation `https://github.com/facebookresearch/moco-v3`. We used torchvision (maintainers & contributors, 2016) implementations. For `a0 (MoCo)`:

- `RandomResizedCrop` with scale=(0.08, 1.0)

Table 6: Effect of architectural and framework adaptations on downstream tasks. We report balanced accuracy for NuCLS (super) and MIDOG25 (majority). Values are mean (s.d.), best values are highlighted in bold.

| Model | NuCLS (super) | MIDOG25 (majority) |
|---|---|---|
| DINOv2-Baseline (ViT-s/8) | 0.681 (0.002) | 0.678 (0.011) |
| DINOv2-No local-to-global (ViT-s/8) | 0.703 (0.003) | 0.673 (0.012) |
| DINOv2-No local-to-global + KDE (ViT-s/8) | 0.767 (0.007) | 0.67 (0.011) |
| MoCov3 (ViT-s/8) | **0.779** (0.005) | **0.746** (0.015) |

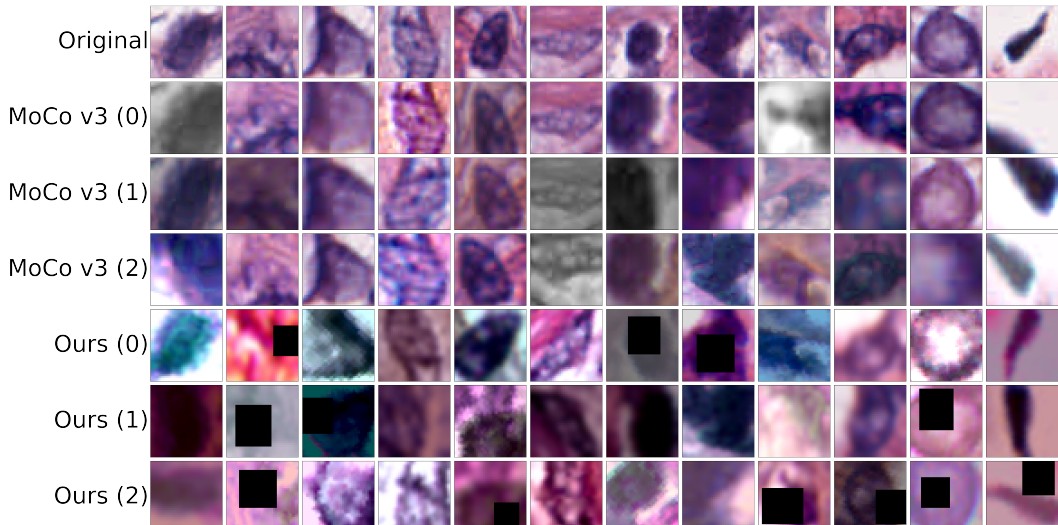

Figure 5: Examples of nuclei images with different augmentation strategies with three random augmentations from either the MoCo v3 augmentation or ours.

- `RandomHorizontalFlip`

- `ColorJitter` with parameters = (0.4, 0.4, 0.2, 0.1) and p=0.8

- `RandomGrayscale` with p=0.2

- `GaussianBlur` with sigma=(0.1, 2.0)

For `a1`:

- `RotationCrop` with degree=360 (our implementation which prevent black pixel padding)

- `RandomResizedCrop` with scale=(0.32, 1.0)

- `RandomHorizontalFlip`

- `ColorJitter` with parameters = (0.6, 0.7, 0.5, 0.2) and p=0.8

- `RandomGrayscale` with p=0.2

- `RandomErasing` with scale=(0.1, 0.3), ratio=(0.8, 1.2) and p=0.3

- `GaussianBlur` with sigma=(0.1, 2.0)

For `a1+gray`, we added `RandomGrayscale` with p=0.2 to the `a1` augmentations. For stain-specific augmentations, we first computed stain statistics on the training dataset with either 1 or 10 mixtures and then sample a new staining to augment the images. `a1-gmm1` and `a1-gmm10` correspond to `a1` with these additional augmenations.

## A.4 Robustness to Staining Variation with Macenko Augmentation

Histopathology images exhibit substantial staining variability across centers, which can induce variations in model embeddings unrelated to true biological differences. To assess the robustness of our models to such variability, we applied Macenko-based stain augmentation and measured the stability of embeddings across reference stains. Six references were selected from TCGA breast samples spanning six centers to capture a wide range of staining characteristics.

We encoded the MIDOG2025 dataset and quantified how the point cloud of embeddings shifts under stain augmentation relative to the original, unaugmented embeddings. Metrics were averaged across the six reference stains. To capture both pointwise and structural differences in the embedding space, we used three metrics: root mean squared error (RMSE) to quantify absolute displacement, cosine similarity to assess directional alignment, and k-nearest neighbor overlap to evaluate the preservation of local neighborhood structure (k=100).

Table 7 reports the stability of embeddings under Macenko stain augmentation for three feature extraction methods: LEMON-MoCov3 (ViT-s/8), Volta (ResNet-18), and H-Optimus-0 (ViT-g/14). LEMON-MoCov3 shows the highest stability across all metrics. Volta exhibits intermediate robustness, while H-Optimus-0 is the most sensitive to staining variation, demonstrating the largest shifts in embeddings and reduced preservation of local neighborhoods.

This difference may be partly explained by the level at which the models operate. LEMON-MoCov3 and Volta are cell-level methods, which focus more on local morphological features of individual cells. Consequently, their embeddings might be less affected by global staining variations. In contrast, H-Optimus-0 is a tile-level model, which captures broader tissue context and features, including staining patterns, potentially making it more sensitive to inter-center variability. These results suggest that cell-level representations are more robust to stain-induced shifts, whereas tile-level embeddings may reflect global staining differences to a greater extent.

This demonstrates that our embeddings capture less staining-related information. To ensure that this robustness does not come at the expense of biologically relevant information, we also evaluated the predictive performance of our models under different staining conditions, comparing results on the MIDOG25 and NuCLS datasets. As shown in Figure 7, LEMON consistently outperforms the competing methods H-optimus-0 and Volta across all augmentation conditions.

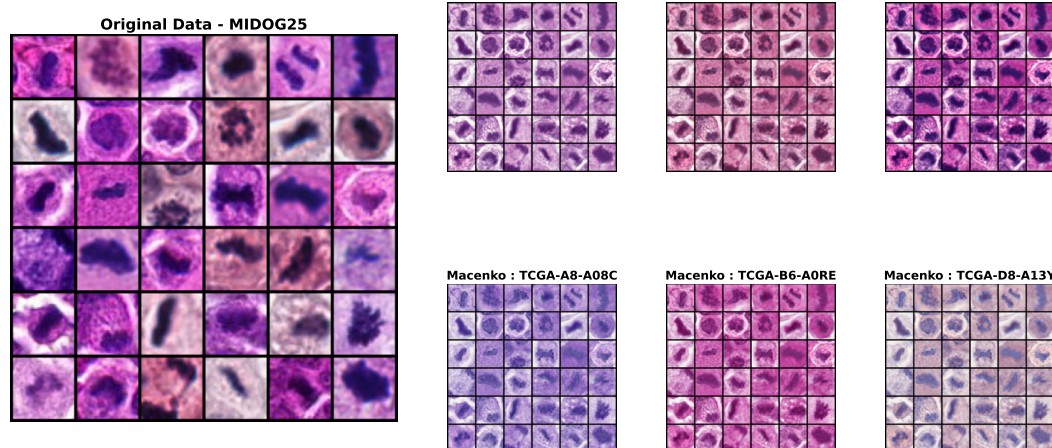

Figure 6: Visualization of cell images under Macenko stain augmentation. The first panel shows a subset of original MIDOG25 cell images, while the remaining panels display the same images after Macenko augmentation using six different reference stains.

Table 7: Robustness of embeddings to Macenko staining variation.

| Metrics | LEMON-MoCov3 (ViT-s/8) | Volta (ResNet-18) | H-Optimus-0 (ViT-g/14) |
|---|---|---|---|
| RMSE (↓) | **0.777** | 0.879 | 1.311 |
| Cosine Similariy (↑) | **0.687** | 0.579 | 0.138 |
| Neightborhood Overlap (↑) | **0.398** | 0.297 | 0.154 |

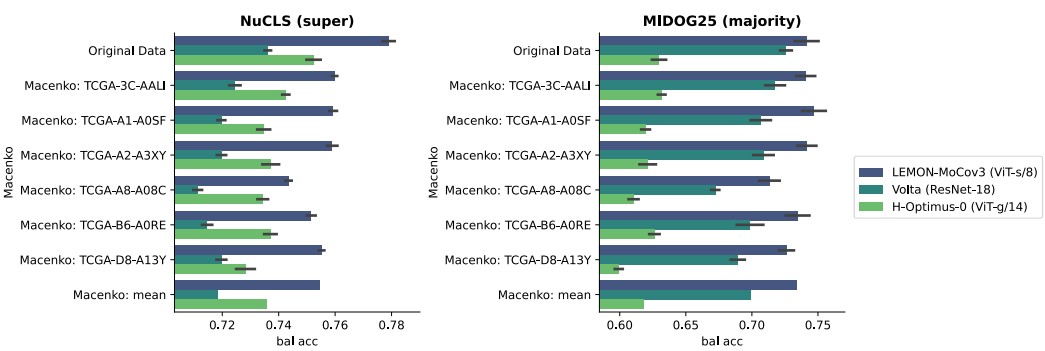

Figure 7: Benchmark of performance under different Macenko staining augmentations. Balanced accuracy of three models (H-optimus-0, LEMON, and Volta) on the NuCLS (left) and MIDOG25 (right) datasets. For each dataset, results are shown for the original images, six Macenko-stained augmentation conditions, as well as the mean across these six augmentations.

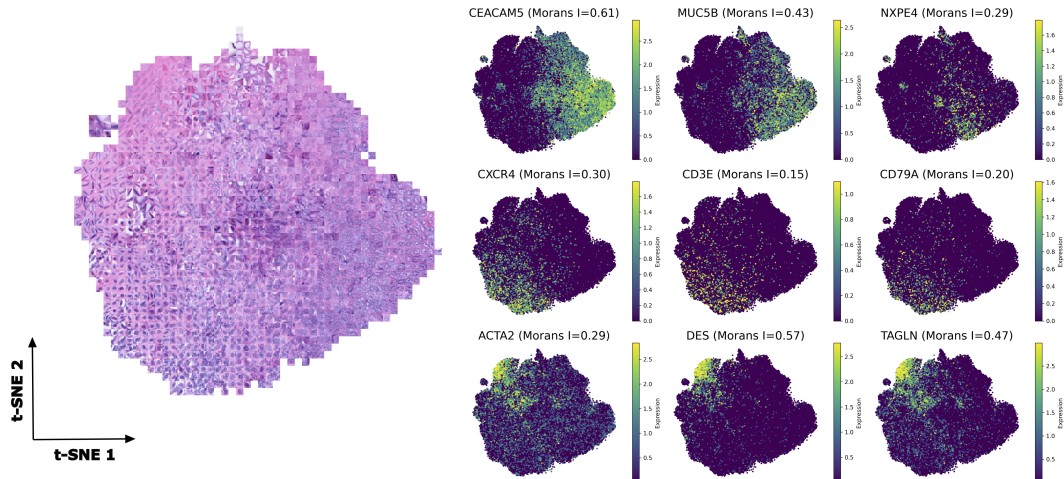

Figure 8: Alignment between morphological embeddings and marker-gene expression in a bowel Xenium tissue section (TENX147). Left: t-SNE projection of cell-level morphological embeddings. Right: normalized expression maps for lineage-associated marker genes, showing their spatial distribution within the morphology-derived manifold.

## A.5 Linking Morphological Embeddings and Gene Expression Patterns

To further investigate whether the morphological embedding space captures biologically meaningful variation in cell state, we performed an additional analysis on a Xenium tissue section. Specifically, we examined the correspondence between (i) the unsupervised structure revealed by the t-SNE projection of the image-derived cell embeddings and (ii) the spatial distribution of expression levels for a curated set of lineage-associated genes representative of epithelial, stromal, and immune identities. To quantify this relationship, we computed Moran's I on the k-nearest-neighbour graph defined in the embedding space. Moran's I captures the degree of autocorrelation of expression values across neighbouring points in the latent manifold.

Figure 8 shows a t-SNE projection of cell embeddings from a bowel Xenium slide, together with expression maps for lineage-associated marker genes spanning epithelial, stromal, and immune identities. Gene expression localizes to defined regions of the morphological manifold, and genes of the same lineage display similar spatial distributions. All markers exhibit positive Moran's I values, indicating that transcriptionally related cells cluster together in the morphology-derived embedding.

## A.6 Comparison of cell embedding strategies from tile-level Foundation Models

When comparing LEMON to tile-level FM, we had to resize nuclei images from $40$ pixels to $224$ pixels, which might be far from the training image distribution of tile-level FM. To investigate whether this was the reason for the mitigated results of these models on our benchmarks, we designed an experiment on the MIDOG25 dataset. For nuclei in this dataset, we extracted crops of size $128 \times 128$ pixels at 40X magnification, centered on each nucleus, which were then resized to $224 \times 224$ to match the expected image size. We selected a crop size of $128 \times 128$ pixels to avoid border effects, as the MIDOG25 dataset did not consist of full WSI. This size therefore allowed us to maximize the number of usable nuclei while remaining compatible with the image distribution used to train the tile-level FM models. With this setting, we retained $9115$ nuclei out of the original $11939$. The label distribution can be seen in Table 8.

To obtain embeddings from tile-level FM, we investigated two approaches to derive the embeddings used in the downstream task. In the **class token** approach, we used the class token embedding as the representation. In the **center tokens** approach, we used the mean embedding of the $4 \times 4$ centered patch tokens of the full image. With our setting, these patches correspond to roughly

$8~\mu m \times 8~\mu m$ in the center of the images, which should contain all the relevant information for the mitosis classification task.

We report the results in Table 9. For all metrics and all tile-level FM, we observe that using the centered patch tokens yields better performance than the class token approach. Our model outperforms all competing approaches, which confirms the relevance of our cell-level FM for cell-level tasks.

| Class | Number of images |
|-------|------------------|
| NMF   | 7955             |
| AMF   | 1160             |

Table 8: MIDOG25 dataset with $128 \times 128$ crop images.

| Target
Metric
Embeddings | MIDOG25 (majority) | | | | |
|---|---|---|---|---|---|
| | auc | aupr | bal acc | f1 (macro) | f1 (weighted) |
| **UNI - class tokens (ViT-l/16)** | 0.729 (0.011) | 0.94 (0.002) | 0.539 (0.007) | 0.543 (0.011) | 0.834 (0.004) |
| **UNI - center tokens (ViT-l/16)** | 0.825 (0.019) | 0.963 (0.006) | 0.645 (0.019) | 0.684 (0.021) | 0.876 (0.007) |
| **Virchow2 - class tokens (ViT-h/14)** | 0.723 (0.019) | 0.937 (0.006) | 0.538 (0.005) | 0.54 (0.009) | 0.833 (0.002) |
| **Virchow2 - center tokens (ViT-h/14)** | 0.85 (0.008) | 0.968 (0.0) | 0.673 (0.017) | 0.714 (0.018) | 0.886 (0.006) |
| **H-Optimus-0 - class tokens (ViT-g/14)** | 0.697 (0.017) | 0.932 (0.006) | 0.524 (0.009) | 0.516 (0.016) | 0.826 (0.005) |
| **H-Optimus-0 - center tokens (ViT-g/14)** | 0.838 (0.02) | 0.966 (0.005) | 0.644 (0.011) | 0.683 (0.012) | 0.875 (0.004) |
| **LEMON-MoCov3 (ViT-s/8)** | **0.912 (0.008)** | **0.984 (0.002)** | **0.727 (0.02)** | **0.768 (0.019)** | **0.905 (0.007)** |

Table 9: Evaluation on the MIDOG25 dataset with $128 \times 128$

## A.7 ADDITIONAL CLASSIFICATION METRICS FOR BENCHMARKS

| Target
Metric
Embeddings | NuCLS (super) | | | | |
|---|---|---|---|---|---|
| | auc | aupr | bal acc | f1 (macro) | f1 (weighted) |
| **Imagenet (ResNet-18)** | 0.876 (0.003) | 0.757 (0.005) | 0.693 (0.005) | 0.698 (0.006) | 0.74 (0.005) |
| **Imagenet (ResNet-50)** | 0.891 (0.004) | 0.777 (0.006) | 0.711 (0.005) | 0.717 (0.005) | 0.756 (0.004) |
| **DINOv2 (ViT-s/14)** | 0.905 (0.001) | 0.794 (0.002) | 0.732 (0.003) | 0.738 (0.004) | 0.775 (0.004) |
| **DINOv2 (ViT-b/14)** | 0.903 (0.003) | 0.792 (0.006) | 0.732 (0.003) | 0.738 (0.004) | 0.775 (0.003) |
| **DINOv2 (ViT-l/14)** | 0.897 (0.003) | 0.775 (0.005) | 0.721 (0.004) | 0.726 (0.005) | 0.767 (0.003) |
| **DINOv2 (ViT-g/14)** | 0.905 (0.004) | 0.786 (0.007) | 0.733 (0.008) | 0.739 (0.008) | 0.777 (0.007) |
| **UNI (ViT-l/16)** | 0.905 (0.004) | 0.789 (0.006) | 0.733 (0.007) | 0.738 (0.007) | 0.777 (0.006) |
| **Virchow2 (ViT-h/14)** | 0.915 (0.004) | 0.807 (0.007) | 0.753 (0.008) | 0.759 (0.008) | 0.794 (0.007) |
| **H-Optimus-0 (ViT-g/14)** | 0.917 (0.002) | 0.812 (0.003) | 0.752 (0.006) | 0.757 (0.007) | 0.794 (0.006) |
| **Volta (ResNet-18)** | 0.905 (0.003) | 0.795 (0.002) | 0.736 (0.003) | 0.74 (0.003) | 0.776 (0.002) |
| **LEMON-MoCov3 (ViT-s/8)** | **0.932 (0.003)** | **0.831 (0.004)** | **0.779 (0.005)** | **0.785 (0.005)** | **0.816 (0.005)** |

Table 10: Evaluation on the NuCLS dataset - task super

## A.8 USE OF LARGE LANGUAGE MODELS (LLMS)

We used LLMs to aid and polish writing.

| Target | MIDOG25 (majority) | | | | |
|---|---|---|---|---|---|
| Metric | auc | aupr | bal acc | f1 (macro) | f1 (weighted) |
| Embeddings | | | | | |
| **Imagenet (ResNet-18)** | 0.781 (0.01) | 0.948 (0.001) | 0.577 (0.008) | 0.597 (0.012) | 0.829 (0.004) |
| **Imagenet (ResNet-50)** | 0.805 (0.014) | 0.955 (0.003) | 0.603 (0.012) | 0.632 (0.016) | 0.84 (0.006) |
| **DINOv2 (ViT-s/14)** | 0.832 (0.011) | 0.961 (0.003) | 0.643 (0.004) | 0.677 (0.004) | 0.856 (0.002) |
| **DINOv2 (ViT-b/14)** | 0.842 (0.007) | 0.964 (0.001) | 0.653 (0.013) | 0.688 (0.013) | 0.86 (0.004) |
| **DINOv2 (ViT-l/14)** | 0.833 (0.012) | 0.961 (0.003) | 0.647 (0.01) | 0.682 (0.009) | 0.858 (0.003) |
| **DINOv2 (ViT-g/14)** | 0.844 (0.01) | 0.964 (0.003) | 0.654 (0.01) | 0.69 (0.009) | 0.861 (0.003) |
| **UNI (ViT-l/16)** | 0.804 (0.012) | 0.953 (0.005) | 0.623 (0.008) | 0.656 (0.01) | 0.849 (0.005) |
| **Virchow2 (ViT-h/14)** | 0.799 (0.013) | 0.95 (0.006) | 0.618 (0.016) | 0.65 (0.02) | 0.848 (0.007) |
| **H-Optimus-0 (ViT-g/14)** | 0.812 (0.011) | 0.954 (0.003) | 0.631 (0.012) | 0.665 (0.015) | 0.852 (0.006) |
| **Volta (ResNet-18)** | 0.849 (0.014) | 0.963 (0.005) | 0.726 (0.01) | 0.742 (0.006) | 0.876 (0.004) |
| **LEMON-MoCov3 (ViT-s/8)** | **0.911 (0.01)** | **0.981 (0.003)** | **0.741 (0.02)** | **0.778 (0.019)** | **0.897 (0.008)** |

Table 11: Evaluation on the MIDOG25 dataset

| Target | PanNuke (classification) | | | | |
|---|---|---|---|---|---|
| Metric | auc | aupr | bal acc | f1 (macro) | f1 (weighted) |
| Embeddings | | | | | |
| **Imagenet (ResNet-18)** | 0.873 (0.002) | 0.589 (0.007) | 0.56 (0.004) | 0.583 (0.005) | 0.643 (0.003) |
| **Imagenet (ResNet-50)** | 0.885 (0.002) | 0.622 (0.005) | 0.587 (0.005) | 0.608 (0.005) | 0.663 (0.003) |
| **DINOv2 (ViT-s/14)** | 0.904 (0.002) | 0.661 (0.006) | 0.635 (0.009) | 0.653 (0.008) | 0.7 (0.003) |
| **DINOv2 (ViT-b/14)** | 0.901 (0.001) | 0.655 (0.005) | 0.627 (0.01) | 0.646 (0.008) | 0.691 (0.003) |
| **DINOv2 (ViT-l/14)** | 0.899 (0.003) | 0.644 (0.009) | 0.619 (0.008) | 0.639 (0.007) | 0.691 (0.003) |
| **DINOv2 (ViT-g/14)** | 0.904 (0.002) | 0.652 (0.006) | 0.631 (0.007) | 0.651 (0.006) | 0.701 (0.002) |
| **UNI (ViT-l/16)** | 0.91 (0.002) | 0.689 (0.005) | 0.643 (0.008) | 0.663 (0.007) | 0.704 (0.003) |
| **Virchow2 (ViT-h/14)** | 0.914 (0.002) | 0.685 (0.007) | 0.645 (0.009) | **0.666 (0.008)** | 0.708 (0.004) |
| **H-Optimus-0 (ViT-g/14)** | **0.915 (0.002)** | **0.691 (0.005)** | **0.648 (0.007)** | 0.665 (0.006) | 0.71 (0.002) |
| **Volta (ResNet-18)** | 0.876 (0.002) | 0.565 (0.002) | 0.571 (0.007) | 0.581 (0.007) | 0.669 (0.002) |
| **LEMON-MoCov3 (ViT-s/8)** | 0.912 (0.001) | 0.643 (0.005) | 0.641 (0.008) | 0.66 (0.007) | **0.712 (0.003)** |

Table 12: Evaluation on the PanNuke dataset