# OpenReview forum: "LEMON - a foundation model for single-cell nuclear morphologies for digital pathology"
_ICLR.cc/2026/Conference — Submitted to ICLR 2026_

### Official Review · Reviewer_PwFY · 2025-10-28

**Soundness:** 3
**Presentation:** 2
**Contribution:** 3
**Rating:** 4
**Confidence:** 3

**Summary:**

The paper proposes LEMON, a self-supervised foundation model designed for single-cell nuclear image representation learning in computational pathology. Unlike existing self-supervised learning (SSL) models trained on slide- or patch-level data, LEMON focuses on cell-level representations. The authors curate a massive dataset (several millions of segmented nuclei from TCGA WSIs) and benchmark LEMON on multiple downstream tasks (cell classification, mitosis detection, and gene expression regression). They compare LEMON with both natural-image and histopathology foundation models (e.g., DINOv2, UNI, Virchow2, Volta) and report superior performance and efficiency.

**Strengths:**

+ The paper addresses foundation models at the single-cell level, which is a real gap in computational pathology. Previous works mostly focus on specific domains or limited data. LEMON generalizes this to a broad, cross-tissue, multi-organ setting and defines a reproducible recipe for training such models.

+ The authors benchmark across seven downstream tasks, spanning classification and regression across several datasets (NuCLS, MIDOG++, HEST). The results consistently demonstrate superior performance, particularly against tile-level foundation models and Volta.

+ Adaptation on MoCov3 and DINOv2 to 40x40 images (e.g., removing local-to-global loss, replacing KoLeo regularizer with KDE) has been carried out, and the authors analyze augmentation policies, dataset scale, and diversity effects. The experiments are systematic, including FLOPs comparison and ablation studies, which lend strong empirical validity and evidence.

**Weaknesses:**

- The methodological contributions are mostly engineering adaptations of existing SSL paradigms (MoCov3 and DINOv2) rather than introducing new theoretical insights or architectures. The novelty lies in scale and application domain rather than new algorithmic concepts.

- While the dataset scale is impressive, details on accessibility, cleaning, and balancing are missing. It is unclear whether the dataset or pre-trained weights will be released (or under what license), which may limit reproducibility and community impact.

- The representation interpretability analysis (e.g., t-SNE in Fig. 4) is qualitative. Biological validation or correlations with known phenotypes or molecular subtypes are absent, which would strengthen claims about biological utility and "foundation model" status.

**Questions:**

1. How well does LEMON generalize to slides from different institutions, staining protocols, or scanners? Have you quantified batch effects beyond the discussion section?

2. Will the pre-trained models or the large-scale cell dataset (or at least sampling scripts) be publicly released? Without this, how can the community benchmark against LEMON?

3. In Fig. 3, performance saturates near 1 M cells. Is this due to model capacity, training strategy, or dataset redundancy? Could larger ViT architectures (ViT-B/8, ViT-L/8) yield further gains with appropriate scaling?

---

> ### Author Response · Authors · 2025-11-24
> **Response to Reviewer PwFY**
>
> We would like to sincerely thank Reviewer PwFY for their thorough and constructive evaluation of our work. We carefully considered all of the points raised and have addressed each of them through additional experiments, analyses, and clarifications in the revised manuscript. Please find the detailed answers to the raised concerns and the questions below:
>
> **Answer to W1:**
> We thank reviewer PwFY for this thoughtful observation. We agree that our work does not introduce a fundamentally new SSL architecture, and that it builds upon established paradigms such as MoCo v3 and DINOv2. Our primary contribution instead lies in demonstrating that, when carefully adapted, scaled, and systematically evaluated, existing SSL frameworks can be repurposed to function as effective single-cell foundation models in computational pathology. This has, to our knowledge, not been previously explored at this scale or level of detail, and our article therefore closes a critical gap through representation learning.
>
> Beyond a change in scale alone, our work required a series of non-trivial methodological design choices and adaptations that are specific to this domain, including adjustments to image resolution, data sampling strategies, augmentation policies, loss components, and architectural configurations. Furthermore, we provide a systematic analysis of how these design choices impact performance across tasks and domains. Finally, we will provide both the foundation model itself and the detection dataset as a scientific resource for future research on single-nucleus morphology foundation models.
>
> **Answer to W2:**
> We thank reviewer PwFY for highlighting the importance of this aspect, which we perfectly agree with: we fully support an open-source, open-data strategy. Accordingly, we would like to clarify that upon acceptance, we will release the TCGA cell detection and pre-trained weights along with Hugging Face cards to ensure accessibility and reproducibility for the community.
>
> In addition, we have anonymously uploaded the LEMON model on Hugging Face  (https://huggingface.co/iclr2025-anonymous/LEMON) for the purpose of the review process, and we will publicly release it under our names upon acceptance.
>
> **Answer to W3:**
> We thank reviewer PwFY for the very interesting suggestion. In answer to this comment (and a similar comment by Reviewer Dpnm), we have conducted additional analyses on the expression of individual genes in order to understand to which extent the phenotypic similarities as observed by cell nuclei correlates with expression of particular genes.
>
> In Appendix Section A.5, we compare our morphological embeddings to cell-type associated marker gene expression in a Xenium tissue section. In addition to a visual inspection of the single-cell gene expression patterns, we quantify this correspondence using Moran’s I computed on the k-nearest-neighbor graph of the embeddings. Our results show that transcriptionally related cells tend to cluster together in the morphology-derived latent space. This analysis provides a quantitative link between the learned embeddings and known biological phenotypes, thereby addressing the concern regarding biological interpretability.

---

> ### Author Response · Authors · 2025-11-24
> **Response to Reviewer PwFY (questions)**
>
> **Answer to Q1:**
> We thank the reviewer for the very interesting question. In Appendix Section A.4, we specifically assess the robustness of LEMON to staining variation across institutions using Macenko-based stain augmentation. We quantify embedding stability across six reference stains from TCGA breast samples (six different centers) using RMSE, cosine similarity, and k-nearest neighbor overlap. LEMON-MoCov3 consistently showed the highest stability compared to other models, demonstrating that its embeddings are robust to batch effects induced by staining variation. Furthermore, we verified that this robustness does not compromise biological relevance by evaluating predictive performance across the MIDOG25 and NuCLS datasets, where LEMON consistently outperformed competing methods under all augmentation conditions. These results indicate that LEMON generalizes well to slides from different centers and staining protocols.
>
> **Answer to Q2:**
> We thank the reviewer for raising this point. Upon acceptance, we will publicly release the LEMON dataset (ie. the full set of segmented nuclei). The original whole-slide images come from the publicly available TCGA dataset. Our release will provide all necessary resources for reproducibility, along with models on Hugging Face.
> In addition, we have anonymously uploaded our best LEMON model on Hugging Face  (https://huggingface.co/iclr2025-anonymous/LEMON) for the purpose of the review process, and we will publicly release it under our names upon acceptance.
>
> **Answer to Q3:**
> We thank the reviewer for this interesting question. As shown in Figure 3, increasing model capacity from ViT-s/8 to ViT-b/8 does not yield a consistent improvement on all datasets, suggesting that model capacity is not the primary factor behind the saturation observed near 1 M cells. We believe this is partly due to the nature of our data: nuclei crops are extremely small images (40×40), which limits the amount of spatial and textural information a larger architecture could exploit. In addition, the morphological diversity of nuclei is inherently lower than that of natural images, further reducing the potential benefits of scaling the backbone.
>
> We believe that diversity is a key factor for the performance of foundation models. This is also suggested by the results presented in Figure 3. Our pretraining set comprises 10,000 WSIs, which is relatively modest compared to tile-level foundation models. The saturation after 1M cells we observe may stem from limited morphological diversity within individual slides rather than the absolute number of cells. Increasing the number of slides could potentially increase the diversity of cells in the pretraining dataset and could therefore further improve performance with larger absolute number of cells.

---

> ### Author Response · Authors · 2025-11-27
> **Feedback**
>
> We would like to kindly thank Reviewer PwFY again for the thoughtful and constructive feedback. As a brief follow-up, we wanted to note that we have addressed all the points raised in our detailed response and revised manuscript, including additional experiments, analyses, and clarifications. We would be very grateful if you had the opportunity to take another look when convenient.

---

### Official Review · Reviewer_Dpnm · 2025-11-01

**Soundness:** 2
**Presentation:** 2
**Contribution:** 2
**Rating:** 4
**Confidence:** 4

**Summary:**

This paper introduces LEMON, a self-supervised foundation model for single-cell nuclear morphology, addressing a gap left by patch- and slide-level models in computational pathology. By curating a large-scale dataset of millions of 40x40 nucleus images from 10 cancer types, the authors adapt and train SSL frameworks for this new domain.

**Strengths:**

- The "Performance vs. FLOPs" analysis (Figure 2) is a nice result, showing that LEMON achieves state-of-the-art performance while requiring orders of magnitude fewer FLOPs than the large tile-level foundation models.

**Weaknesses:**

- The finding that MoCov3 consistently outperforms the modified DINOv2 (Table 5) is a key result. The paper hypothesizes this is due to the 40x40 image size favoring a global contrastive objective. However, the DINOv2/iBOT framework includes a masked-image modeling objective specifically designed to learn fine-grained local features. Why do the authors believe this objective, which seems ideal for capturing chromatin-level detail, fails in this low-resolution setting?
- The dataset composition study (Figure 3, top row) shows performance saturating at just 1M cell images, yet the final models are trained on 10M cells. Why was 10M used if saturation occurred at 1M? Furthermore, this 1M saturation point seems surprisingly low for an SSL model. Does this suggest that the morphological diversity of nuclei is relatively constrained, and the model learns the full distribution of appearances after only 1M examples?
- The comparison to tile-level foundation models (UNI, Virchow2) in Table 1 involved resizing the 40x40 cell images up to 224x224 to match the models' expected input size. This resizing likely introduces significant artifacts and is far from the models' training distribution. How confident are the authors that the poor performance of these models isn't just an artifact of this "up-resizing" protocol, rather than a true failure of tile-level features to capture cell morphology?

**Questions:**

- The augmentation study (Table 2) is excellent. It's surprising that a "realistic" stain augmentation like RandstainNA (a1+gmm1) hurt classification performance compared to the a1+gray policy. Why do the authors think a more realistic simulation of stain variation was detrimental for classification tasks, while a simple grayscale augmentation was beneficial?
- The t-SNE analysis in Figure 4 provides strong qualitative evidence that the embedding space captures morphological phenotypes, which are annotated by a pathologist . The model is also benchmarked on gene expression prediction (HEST). Is there a way to bridge these two results? For example, if the cells in the t-SNE are colored by their ground-truth (or predicted) expression of a known marker gene, does this align with the expert-annotated morphological clusters?

---

> ### Author Response · Authors · 2025-11-24
> **Response to reviewer Dpnm**
>
> We would like to sincerely thank Reviewer Dpnm for their thorough and constructive evaluation of our work. We carefully considered all of the points raised and have addressed each of them through additional experiments, analyses, and clarifications in the revised manuscript. Please find the detailed answers to the raised concerns and the questions below:
>
> **Answer to W1:**
> We thank the reviewer for this interesting comment. First, we observe empirically that MoCo v3 consistently achieves better performance than DINOv2. We then attempted to identify possible reasons for this result, especially given that DINOv2 is among the strongest existing SSL frameworks. Our hypothesis is that DINOv2 may simply be less effective at this low spatial resolution. While masked-image modeling is indeed an effective objective for learning fine-grained local features, one of the strengths of the DINOv2 framework lies in its local-to-global matching. In our setting, however, the “local” regions may be too small to provide meaningful information. If this local-to-global matching component is weakened or removed, the overall effectiveness of the framework may be compromised, even if the masked-image modeling objective still contributes useful signals. Another possible explanation is that chromatin-level details are too localized to be reliably inferred from the surrounding context in such low-resolution patches.
>
> **Answer to W2:**
> We chose the model pretrained on the largest number of images, as models exposed to more diverse datasets are generally expected to exhibit better generalization. In our specific experiments, it is possible that this choice does not substantially change the results; however, since the performance is at least on par with the alternatives, we consider it preferable to rely on this model. Importantly, this choice does not entail longer training times, as we define one epoch as processing 1M cell images.
>
> Regarding the pretraining data, we believe that diversity is the key factor, as also suggested by the experiments in Figure 3. Our pretraining set comprises 10,000 WSIs, which is relatively modest compared to tile-level foundation models. The saturation after 1M cells we observe may stem from limited morphological diversity within individual slides rather than the absolute number of cells. Increasing the number of slides could potentially increase the diversity of cells in the pretraining dataset and could therefore further improve performance with larger absolute number of cells.
>
> Another possible explanation, in line with the idea proposed by the reviewer, is that the morphological diversity of nuclei is limited compared to that of natural images. An additional contributing factor may be the relatively small image size, which further reduces the expected diversity, as there are fewer degrees of freedom, simply because there are fewer pixels to encode this variability.
>
> **Answer to W3:**
> We thank the reviewer for this insightful comment and we agree with this concern. Indeed, resizing 40×40 cell images to 224×224 to match tile-level foundation models such as UNI and Virchow2 is far from their original training distribution and may introduce artifacts that negatively impact performance. We have now explicitly acknowledged this limitation in the revised manuscript. On the other hand, this procedure has been proposed in recent work on single-cell classification (Banerjee et al., 2025). At the same time, we believe this further highlights the need for dedicated cell-level foundation models that operate directly at the appropriate resolution, which is the motivation of this work.
>
> Banerjee, S.; Weiss, V.; Donovan, T. A.; Fick, R. H. J.; Conrad, T.; Ammeling, J.; Porsche, N.; Klopfleisch, R.; Kaltenecker, C.; Breininger, K.; Aubreville, M.; Bertram, C. A. Benchmarking Deep Learning and Vision Foundation Models for Atypical vs. Normal Mitosis Classification with Cross-Dataset Evaluation. arXiv July 16, 2025. https://doi.org/10.48550/arXiv.2506.21444.

---

> ### Author Response · Authors · 2025-11-24
> **Response to Reviewer Dpnm (questions)**
>
> **Answer to Q1:**
> We thank the reviewer for the positive assessment of our augmentation study and the insightful question. Regarding the augmentation study (Table 2), we observed that the “realistic” RandstainNA (a1+gmm1) sometimes decreased classification performance compared to the simpler a1+gray policy. We hypothesize that this is because a1+gray encourages the model to focus on local morphological features rather than global color variations when learning to recognize positive pairs. By drastically reducing color information, the model is forced to rely on cell morphology, which provides the most informative signal for cell-level classification and also makes the embeddings more robust to staining variation. This robustness is particularly advantageous on datasets like MIDOG25, where domain shifts between centers and staining protocols present a significant challenge.
>
> **Answer to Q2:**
> We thank the reviewer for this excellent suggestion. In Appendix Section A.5, we explicitly investigated the biological relevance of our morphological embeddings by comparing them to lineage-associated gene expression in a Xenium tissue section. We quantified this correspondence using Moran’s I on the k-nearest-neighbour graph of the embeddings, showing that cells which cluster together in the morphology-derived latent space also have similar transcriptomic state. This analysis provides a quantitative link between the embeddings and known biological phenotypes, addressing the concern regarding biological interpretability.

---

> ### Author Response · Authors · 2025-11-27
> **Feedback**
>
> We would like to kindly thank Reviewer Dpnm again for the thoughtful and constructive feedback. As a brief follow-up, we wanted to note that we have addressed all the points raised in our detailed response and revised manuscript, including additional experiments, analyses, and clarifications. We would be very grateful if you had the opportunity to take another look when convenient.

---

### Official Review · Reviewer_RDwH · 2025-11-06

**Soundness:** 3
**Presentation:** 3
**Contribution:** 3
**Rating:** 6
**Confidence:** 5

**Summary:**

The authors introduce a novel family of vision foundation models called LEMON for digital pathology operating at the cell level. They propose to first extract images of 40*40 pixels of each cell within an image, before learning cell representations using well-known self-supervised learning frameworks such as MoCov3 and DinoV2. To this end, they investigate various ViT architectures, SSL losses and data augmentation strategies, while pre-training models on several million of cells extracted from around 10k WSI of the TCGA database. The authors show that their best model (relying on MoCov3) outperforms a range of SSL models, learned on either natural or pathology images, on 5 cell-level tasks. Then they provide a range of ablation studies on the aforementioned different SSL settings and study the scaling laws of their LEMON-MoCov3 model.

**Strengths:**

- Overall, the paper is well-written.
- Propose a novel family of vision foundation models for cell-level tasks pre-trained on 10M cell images extracted from the large-scale TCGA database.
- Study various SSL frameworks including MoCoV3, DinoV2 and variants of the latter, where they observe that its local-to-global objective is detrimental to learning cell representations.
- Study various augmentation strategies with a concern for batch effect mitigation.
- Show that Lemon-mocov3 outperforms many baselines over 4 cell-level tasks with a significant margin, while being competitive with the best baselines on a 5th task.
- Study scaling laws for Lemon-mocov3, showing that performance tends to saturate over 1M cell images.

**Weaknesses:**

- **W1: removing spatial information between cells**. Models proposed by the authors inherently remove spatial/environmental information between cells, which is an important factor in digital pathology, e.g within tertiary lymphoid structures to discriminate different immune cells [A, B]. I remark that Volta includes environmental information and achieves quite competitive performances while using a likely weaker encoder and learned from less data. I believe that it would be interesting to further test the importance of that information. For instance, as we can observe fairly weak performances of tile-level pathology FM on the cell-level task, which I believe is because zooming in like authors did to get cell embeddings lead to a too big domain drift for these models, some spatially-informed baselines could be derived from these model I suggest a rather simple baseline which assumes that these models embed patch tokens (14*14 pixels) and image CLS token into an Euclidean space. One could compute a tile embedding and corresponding cell-segmentation mask. Then, for each cell, a cell embedding could be computed as a convex combination of the patch token embeddings it belongs to, whose weights coincide with the proportions of pixels associated with the cell contained in the patch token.


[A] Pitzalis, C., Jones, G. W., Bombardieri, M., & Jones, S. A. (2014). Ectopic lymphoid-like structures in infection, cancer and autoimmunity. Nature Reviews Immunology, 14(7), 447-462.

[B] Schaadt, N. S., Schönmeyer, R., Forestier, G., Brieu, N., Braubach, P., Nekolla, K., ... & Feuerhake, F. (2020). Graph-based description of tertiary lymphoid organs at single-cell level. PLoS Computational Biology, 16(2), e1007385.

- **W2: data curation**. It seems to be an overstatement in Section 2.1 that a data curation technique was applied, while the authors simply randomly sample cells within WSI if I understood correctly. Methods such as the one in Vo & al (2024), also studied recently for histopathology FM in [C] with tailored batch sampling strategies, could have been relevant to use in the paper.

[C]  Chen, B., Vincent-Cuaz, C., Schoenpflug, L. A., Madeira, M., Fournier, L., Subramanian, V., ... & Rätsch, G. (2025, September). Revisiting Automatic Data Curation for Vision Foundation Models in Digital Pathology. In International Conference on Medical Image Computing and Computer-Assisted Intervention (pp. 554-564). Cham: Springer Nature Switzerland.

- **W3: clarity**. It can be beneficial for the paper to clarify certain points, including the following
     - a) The choice of SSL methods included in the study, namely MoCoV3 and DinoV2, is not clearly justified. Other methods, such as MAE, BeiT etc, could have been considered, knowing that the authors observed that DinoV2 was not performing that well within the LEMON framework. This matter should be discussed in the paper and therefore the distinction between SSL methods should be more detailed than the current one, being contrastive methods vs non-contrastive ones.
     - b) In section 2.1 can you mention in the paper at which magnification the TCGA images have been processed, knowing that it includes both 20x and 40x images?
     - c) In Figure 3, could the authors had a last column with global performance computed as the averaged performance across datasets to ease comparison between configurations?


- **W4: benchmark**. I believe that some points should be added/clarified in the current benchmark
      - a) It is not clear in this type of applications why authors used a balanced accuracy to compare methods instead of e.g a macro f1 score or AUPRC. Could you provide these complementary metrics ?
      - b) Moreover it seems to be important to underline that NuCLS comes from TCGA WSIs hence is an in-domain dataset. Finally, I believe that it would be relevant to include the PanNuke dataset which is a large dataset often used in the literature (see CellVIT and so on).

- **W5. batch effect**. As suggested by the authors in their augmentation schemes and conclusion, batch effects are important to consider. It could be relevant to add to the paper an evaluation on that matter and also to consider benchmarks taking transformed images (e.g using Macenko or variants) as inputs instead of raw images.

**Questions:**

I invite the authors to address/discuss the weaknesses mentioned above which are already mixed with questions for conciseness.

---

> ### Author Response · Authors · 2025-11-24
> **Response to Reviewer RDwH**
>
> First, we would like to thank Reviewer RDwH for the positive assessment of our manuscript, as well as for the constructive and insightful comments and suggestions. Please find below our responses to all the concerns raised, including a description of the additional experiments we conducted.
>
> **Answer to W1:**
> We fully agree that incorporating spatial context is an important direction for modeling cellular organization, tissue architecture, and multi-cellular patterns relevant to disease processes. The key question, however, is when this spatial information should be introduced: directly at the level of the single-cell representation, or only after robust single-cell features have been learned.
>
> Our work deliberately follows the second strategy, i.e., learning spatially agnostic single-cell representations and incorporating spatial relationships in a downstream module if required. We believe this choice offers several advantages. First, it preserves the generality of the representations: many cell-level tasks—such as mitosis detection or nuclear atypia scoring—depend primarily on intrinsic cell morphology rather than neighborhood structure. Embedding spatial context into the feature itself risks reducing versatility by making the representation dependent on the surrounding microenvironment, even for tasks where this is unnecessary or undesirable.
>
> Second, separating morphology from spatial modeling provides a more modular and interpretable pipeline. If neighboring cells directly influence the representation of each individual cell, then any subsequent spatial analysis becomes harder to interpret, and risks a degree of circularity: spatial dependencies affect the representation, and those representations are then used to infer spatial structure. By disentangling the two levels - first learning morphology, then modeling spatial organization - we obtain a cleaner framework where spatial effects can be added, modified, or ablated in a controlled manner.
>
> Finally, we would like to emphasize that our present focus is explicitly on building high-quality morphological cell-level representations. Spatial modeling could operate on LEMON representations, and we consider this a valuable direction for future work.
>
> **Answer to W2:**
> We thank the reviewer for this observation. We agree that the term “curated” may have been misleading in this context, and we also agree that sampling strategies for large-scale cell extraction are an important research question. In fact, we initially explored several structured sampling approaches, including clustering-based techniques and stratified sampling across organs, slides, or morphological prototypes. However, we found that such methods introduce a large number of hyperparameters (e.g., number of clusters, number of samples per cluster, representation used for clustering, number of organs or slides per batch). Selecting these hyperparameters in a principled and unbiased way proved extremely difficult, and we were concerned that any such choices could inadvertently influence downstream analyses.
>  For this reason, we ultimately adopted random sampling, as it provides a neutral and assumption-free strategy that avoids injecting unintended biases into the dataset. Nonetheless, the paper does provide insights related to sampling. In particular, Figure 3 analyzes how performance varies with the number of slides, nuclei, and organs used for training. We believe these observations are valuable for researchers interested in constructing new single-cell foundation models.
>
>  While “curated” was not the ideal word choice, our intention was to emphasize that the dataset creation itself represents an important contribution of our work. We will make the entire collection of cell detections publicly available upon acceptance, and we hope it will serve as a strong starting point for future investigations into sampling strategies, data curation techniques, and large-scale single-cell modeling for computational pathology.

---

> ### Author Response · Authors · 2025-11-24
> **Response to Reviewer RDwH - 2**
>
> **Answer to W3.a:**
> We thank the reviewer for this insightful comment. Our choice to focus on MoCoV3 and DINOv2 was driven by their central role in existing pathology foundation models. Contrastive approaches derived from MoCoV3 form the backbone of several influential tile-level models (e.g., CTransPath), while DINOv2 and its variants are the basis of recent large-scale pathology encoders such as UNI, Virchow and H-Optimus. Given this strong empirical presence in the field, we considered these two families of SSL-methods for our new cell-level foundation model.
>
> Regarding other SSL approaches such as MAE or BEiT, we fully agree that masked image modeling constitutes a promising direction for histology and potentially for cell-level learning as well. We have now given the rationale behind choosing MocoV3 and DinoV2 in the section “Choice of SSL framework”.
>
>
> **Answer to W3.b:**
> We thank the reviewer for pointing out this important clarification. In our experiments, we used exclusively whole slide images scanned at 40× magnification, as this resolution provides the level of detail necessary for capturing fine-grained nuclear morphology, which is essential for cell-level representation learning. We fully agree that this information is crucial for reproducibility, and we have now explicitly added this specification to Section 2.1 in the revised version of the manuscript.
>
> **Answer to W3.c:**
> We agree with the reviewer that a summary graph might be better suited to provide a concise overview of the benchmark results. We have added this overview graph in the revised version.
>
> **Answer to W4.a:**
> We thank the reviewer for this helpful suggestion. We agree that reporting additional metrics provides a more comprehensive view of model performance. In response, we have added AUPRC, AUC, Macro-F1 and weighted F1 scores for all evaluated methods in Appendix. The overall conclusions remain consistent across these complementary metrics. On NuCLS and MIDOG25, LEMON continues to outperform all baselines by a substantial margin. While results are more heterogeneous on PanNuke, LEMON remains among the top-performing models for most metrics.
>
> **Answer to W4.b:**
> We thank the reviewer for this insightful comment. Indeed, NuCLS is an in-domain dataset; however, to the best of our knowledge, this applies to all the models considered. MIDOG25 is probably an out-of-domain dataset for all methods, as it was released after the publication of the foundation models. However, as it was derived from MIDOG++, released in 2023, we cannot be entirely sure about this. Regarding PanNuke, we are also not certain whether it was used to train the competing foundation models. We have clarified this point in the revised manuscript, in Appendix A.1.
>
> **Answer to W5:**
> We thank the reviewer for this valuable suggestion, which we have addressed through additional analyses. Following the reviewer’s advice, we tested how different SSL methods handle domain shifts. To this end, we transformed the cell patches using the Macenko method and report the representation differences induced by this transformation, as measured by RMSE, cosine similarity, and overlap of the k nearest neighbors, in order to assess how well local structures in the latent space are preserved (see Table 6; Appendix section A.4). Furthermore, we evaluated the effect of the Macenko-induced domain shift on downstream performance when using the competing representations and found that LEMON outperforms the competing methods also under domain shift (see Figure 7; Appendix section A.4).

---

> ### Comment · Reviewer_RDwH · 2025-11-25
> **Response to authors**
>
> Thank you for your detailed answers. Overall, I find the rebuttal compelling, except on W1 which I discuss further below. I still encourage authors to clarify which datasets may contain cells seen during LEMON's pre-training (not necessarily those of all competitors) and those which should not as discussed in W4-b.
>
> **Discussion on W1**: I agree with the authors that when spatial information should be introduced remains unclear. However, I still strongly believe that the current cell-level benchmark remains a very good opportunity to check whether including spatial information, or even extra-cellular and connective tissue information, would be relevant for these tasks, which are supposed to be fairly local. I don't consider that it is out of the scope of this paper because pathology foundation models (e.g UNI) seek to be able to learn robust visual features across diverse biological scales ranging from cells to tissues, and also be able to generalize across very diverse resolutions and magnifications. Therefore to some extent, one could see LEMON as a relevant extension of these works, suggesting to introduce in their learning strategies images of cells at very high resolution but low magnification (resulting from zooming in on a cell originally within a 40x WSI). And consequently, it is not clear why models only trained on this type of cell images would be more relevant for digital pathology than such highly generalist models. Therefore, I still strongly encourage authors to evaluate the baselines I mentioned in my first review for their rebuttal.

---

> > ### Author Response · Authors · 2025-11-28
> > **Comparison of cell embedding strategies from tile-level Foundation Models**
> >
> > We thank the reviewer for their reactivity and constructive feedback. We clarified in Section 2.3 and Appendix 1.A which datasets are in-domain or out-of-domain with respect to LEMON’s pretraining (W4-b).
> > Discussion on W1. We thank the reviewer for this insightful comment. We agree that applying a tile-level Foundation Model on the resized nucleus crops might suffer from an important domain shift between the tile images on which the FM is trained and the nucleus crops it is applied to (even though this strategy has been used before).
> >
> > To directly address this concern, we therefore added an experiment on the MIDOG25 dataset. Specifically, we extracted 128×128 pixel images at 40× magnification, centered on the nucleus. Then, we evaluated representations obtained with a tile-level FM in two ways: (i) using the embedding from the class token, and (ii) using the mean embedding of the 4×4 centered patches, which approximately correspond to the centered cell image. We selected a crop size of 128×128 pixels to avoid border effects.
> >
> > The results are reported in the section **Comparison of cell embedding strategies from tile-level Foundation Models**. As can be seen there, our approach consistently outperforms these newly introduced baselines. These experiments support the view that pathology FMs should also explicitly consider such high-resolution, cell-centric images in their training objectives to better capture the full range of biological scales.
> >
> > We hope that these clarifications and additional experiments fully address the reviewer’s concerns.

---

### Author Response · Authors · 2025-11-24
**Overall response**

We would like to thank all reviewers for the assessment of our manuscript, as well as for the constructive and insightful comments and suggestions. We have revised our paper and Appendix, uploaded a new version to OpenReview. We have conducted additional experiments, added new analyses, and incorporated clarifications suggested by the reviewers.
Specifically, we have:
- **Expanded augmentation and batch effect analyses:** We quantified the robustness of LEMON embeddings under Macenko stain augmentation across six TCGA reference centers. Metrics including RMSE, cosine similarity, and k-NN overlap demonstrate that LEMON is more robust to staining variation than competing models. We also evaluated downstream classification performances under these augmentations where on MIDOG25 and NuCLS, LEMON demonstrated better performances.
- **Biological interpretability analyses:** We compared single-cell embeddings to lineage-associated gene expression in a Xenium tissue section. Moran’s I on the k-NN graph providing a quantitative link between embeddings and known phenotypes and shows that cells which cluster together in the morphology-derived latent space also have similar transcriptomic state; .
- **Benchmark and metric updates:** We expanded the benchmark with Macro-F1 and AUPRC scores in Appendix and included PanNuke dataset to the benchmark. Overall conclusions remain consistent. On NuCLS and MIDOG25, LEMON continues to outperform all baselines by a substantial margin. While results are more heterogeneous on PanNuke, LEMON remains among the top-performing models for most metrics.
- **Dataset and data accessibility:** We clarified some details such as: all TCGA WSIs were scanned at 40X magnification, and that our dataset comprises randomly sampled nuclei to avoid sampling biases. Upon acceptance, we will release the nuclei detections (coordinates) and pre-trained models to provide access to our model, and a starting point for future studies on single-cell foundation models.
- **Discussion on SSL framework:** We clarified in our responses why we focused on MoCo v3 and DINOv2, distinguishing contrastive versus masked-image modeling approaches, and explaining the observed performance differences in the context of model capacity and dataset diversity.

We believe that these additional analyses and experiments significantly strengthen the paper.

---

> ### Author Response · Authors · 2025-11-28
> **Comparison of cell embedding strategies from tile-level Foundation Models**
>
> We have added additional experiments to investigate different cell embedding strategies from tile-level Foundation Models to assure a fair comparison. For this, we extracted 128×128 pixel images at 40× magnification, centered on the nucleus. Then, we evaluated representations obtained with a tile-level FM in two ways: (i) using the embedding from the class token, and (ii) using the mean embedding of the 4×4 centered patches, which approximately correspond to the centered cell image. We selected a crop size of 128×128 pixels to avoid border effects.
>
> The results are reported in the section **Comparison of cell embedding strategies from tile-level Foundation Models**. As can be seen there, our approach consistently outperforms these newly introduced baselines.
>
> We would like to thank again the reviewers for their constructive comments which - in our view - have significantly improved the paper quality.

---

### Meta-Review · Area_Chair_kivP · 2026-01-06

**Summary:**

This paper proposed LEMON, a self-supervised foundation model for single-cell image representation. Empirical results show robust morphology representations that benefit large-scale single cell pathology studies. One reviewer was positive overall while two reviewers were slightly negative due to different reasons. Some major concerns were addressed during the rebuttal (see the detailed list below), but there are some important concerns that remain. At this stage, I found it a boarder line paper and I slightly lean toward rejecting it since some key questions were not directly answered:

1. The authors admit that they over-claimed the contribution of data "curation". Please make the contribution clear and fair.
2. Most questions by Reviewer Dpnm were not really addressed by the reviewer with convincing evidence:
    2.1 why MoCov3 model consistently outperforms DINOv2?
    2.2 why the model saturate at 1m samples?
    2.3 why the results in table 2 are surprising.
    2.4 The potential artifact from resizing is not thoroughly discussed and supported by experiments.
I agree with the reviewer that these are all important points that should be addressed before being published.

**Reviewer Concerns:**

Reviewer RDwH
1. Removing spatial information (addressed by detailed explanation)
2. The use of "curation" (not really addressed. The authors admit that the word is not accurate)
3. The choice of SSL methods (addressed by a new section "Choice of SSL framework")
4. Benchmark concerns (addressed by new experimental results in appendix)
5. batch effect (addressed by new experimental results in Appendix A.4)

Reviewer Dpnm
1. Reason for the superior performance of MoCov3 over DINOv2 (partially addressed by some hypothesis)
2. Model saturation (partially addressed by some heuristic explanation)
3. Potential artifact from resizing (not addressed)
4. Reason behind results in Table 2 (partially addressed by some hypothesis)
5. Link between embeddings and gene expression  (not directly addressed, but authors provided other experiments to  link embeddings with known phenotypes).

Reviewer PwFY
1. Limited novelty in terms of theoretical insights or architectures (partially addressed by clarifying the contribution)
2. Lack of details about accessibility of the proposed method (addressed by new Huggingface model and promised release of model and weights on Huggingface)
3. Lack of quantitative evaluation of the representation interpretability analysis (addressed by additional experiments in Appendix A.5)

**Reviewer Scores:**

Reviewer RDwH: 6 --> 6

Reviewer Dpnm: 4 --> 4

Reviewer PwFY: 4 --> 5

---

### Decision · Program_Chairs · 2026-01-26

Reject